Aggressive nutrition in extremely low birth weight infants: impact on parenteral nutrition associated cholestasis and growth

Repa Andreas andreas.repa@meduniwien.ac.at 1
Lochmann Ruth 1
Unterasinger Lukas 1
Weber Michael 2
Berger Angelika 1
Haiden Nadja 1
1 Department of Pediatrics and Adolescent Medicine, Division of Neonatology, Pediatric Intensive Care and Neuropediatrics, Medical University of Vienna , Vienna , Austria
2 Department of Radiology, Medical University of Vienna , Vienna , Austria
Meyre David
Electronic publication date: 2016 Sep 20
Publication date: 2016
Volume: 4
Electronic Location ID: e2483
Received 2016 Jun 5; Accepted 2016 Aug 24
Copyright: ©2016 Repa et al.
Copyright year: 2016
Copyright holder: Repa et al.
License: This is an open access article distributed under the terms of the Creative Commons Attribution License, which permits unrestricted use, distribution, reproduction and adaptation in any medium and for any purpose provided that it is properly attributed. For attribution, the original author(s), title, publication source (PeerJ) and either DOI or URL of the article must be cited.
License URL: https://creativecommons.org/licenses/by/4.0/

Keywords: Extremely low birth weight infants, Parenteral nutrition, Growth, Cholestasis, Aggressive nutrition

Funding: The authors received no funding for this work.

==============================
Background

Parenteral nutrition associated cholestasis (PNAC) is a frequently observed pathology in extremely low birth weight (ELBW) infants. Its pathogenesis is determined by the composition and duration of parenteral nutrition (PN) as well as the tolerance of enteral feeds (EF). “Aggressive” nutrition is increasingly used in ELBW infants to improve postnatal growth. Little is known about the effect of “aggressive” nutrition on the incidence of PNAC. We analyzed the influence of implementing an “aggressive” nutritional regimen on the incidence of PNAC and growth in a cohort of ELBW infants.

Methods

ELBW infants were nourished using a “conservative” (2005–6; n = 77) or “aggressive” (2007–9; n = 85) nutritional regimen that differed in the composition of PN after birth as well as the composition and timing of advancement of EFs. We analyzed the incidence of PNAC (conjugated bilirubin > 1.5 mg/dl (25 µmol/l)) corrected for confounders of cholestasis (i.e., NEC and/or gastrointestinal surgery, sepsis, birth weight, Z-score of birth weight, time on PN and male sex), growth until discharge (as the most important secondary outcome) and neonatal morbidities.

Results

The incidence of PNAC was significantly lower during the period of “aggressive” vs. “conservative “nutrition (27% vs. 46%, P < 0.05; adjusted OR 0.275 [0.116–0.651], P < 0.01). Body weight (+411g), head circumference (+1 cm) and length (+1 cm) at discharge were significantly higher. Extra-uterine growth failure (defined as a Z-score difference from birth to discharge lower than −1) was significantly reduced for body weight (85% vs. 35%), head circumference (77% vs. 45%) and length (85% vs. 65%) (P < 0.05). The body mass index (BMI) at discharge was significantly higher (11.1 vs. 12.4) using “aggressive” nutrition and growth became more proportionate with significantly less infants being discharged below the 10th BMI percentile (44% vs. 9%), while the percentage of infants discharged over the 90th BMI percentile (3% vs. 5%) did not significantly increase.

Discussion

“Aggressive” nutrition of ELBW infants was associated with a significant decrease of PNAC and marked improvement of postnatal growth.

Introduction

Parenteral nutrition associated cholestasis (PNAC) is the most common form of liver damage in neonates while receiving parenteral nutrition (PN) (Kelly, 1998). Its prevalence was reported between 10% and 60% in neonatal intensive care units (Klein et al., 2010). Preterm infants born <1,000 g (extremely low birth weight (ELBW) infants) are at particular risk to develop PNAC due to a prolonged dependence on PN, low tolerance of enteral feedings (EF), immaturity of hepato-biliary function and infections (Beale et al., 1979; Beath et al., 1996; Drongowski & Coran, 1989). With regard to PN, high cumulative doses of amino acids (Steinbach et al., 2008) and soybean based lipid emulsions (Carter et al., 2007; Clayton et al., 1993; Kurvinen et al., 2012) were suggested to play an important role in PNAC pathogenesis. To avoid PNAC and other complications associated with PN, parenteral proteins and lipids were hence frequently withheld right after birth unless “metabolic tolerance” was assumed (Pereira, 1995) and EFs were only cautiously introduced in fear of necrotizing enterocolitis (NEC) (Morgan, Young & Mcguire, 2011). However, this “conservative” approach of nutrition caused a deficit in protein and energy supply (Embleton, Pang & Cooke, 2001) leading to postnatal growth failure—a common phenomenon in ELBW infants (Fanaroff et al., 2007). Today, the contemporary nutritional practice has become more “aggressive” (Hay Jr, 2013) in terms of early application of higher amounts of amino acids and lipids in PN right after birth and more rapid progression of EFs with higher caloric density. To reduce growth failure of ELBW infants at our unit, we thus changed our nutritional approach from a “conservative” to a more “aggressive” regimen (Table 1). However, higher amounts of amino acids (Steinbach et al., 2008) and lipids (Carter et al., 2007; Clayton et al., 1993; Kurvinen et al., 2012) in early PN may increase PNAC incidence. On the other hand, earlier intensive enteral nutrition may improve bile flow and prevent PNAC. While regimens of “aggressive” nutrition including both aspects are used in clinical routine more often, their influence on the incidence of PNAC has not been investigated (Guellec et al., 2015). We therefore performed a retrospective observational study with the primary aim to investigate the effect of implementing “aggressive” nutrition on the incidence of PNAC and—as a secondary aim—to evaluate the effects on growth in ELBW infants.

Table 1 Guidelines on nutrition of ELBW infants.

Parenteral	Glucose (g/kg/d)	Protein (g/kg/d)	Lipids (g/kg/d)	
Conservative	6.0 (day 1)–18a	0.5 (day 1)–4.0b	0.5 (day 3)–3.5c	
Aggressive	7.2 (day 1)–18a	2.0 (day 1)–4.0b	1.0 (day 1)–4.0c	
Enteral	First feedings	Breast milk fortifier	Formula	
Conservative	stable infant, no defined increase	at full EF	Start with diluted eHFd; preterm formula at full EF	
Aggressive	first day of life, increase max. 20 ml/kg/d	at 100 ml/kg EF	Always preterm formula	
Notes.

Guidelines for nutrition of ELBW infants using a conservative or aggressive approach. Using aggressive nutrition, parenteral protein and lipids were introduced earlier and EFs were started with full strength preterm formula instead of diluted eHF on the first day of life. Enteral feedings were always started with formula and switched to own mother’s breast milk as soon as available. Breast milk fortification was started earlier using the aggressive approach.

a increased by 1–2 g/kg/d as tolerated.

b increased by 0.5 g/kg/day.

c increased by 0.5 g/kg/day as tolerated.

d started at half strength, increased stepwise to full strength at half enteral nutrition.

EF enteral feeds

eHF extensively hydrolyzed formula

Materials & Methods

Nutritional and clinical management of ELBW infants

The nutritional guidelines for ELBW infants at our unit are shown in Table 1.

With “conservative” nutrition, PN was started after birth with glucose only, while protein and fat were slowly introduced during the first week of life. Enteral feedings were started using an extensively hydrolyzed formula (eHF) at half strength (10 kcal/oz) when the infant was clinically stable. As soon as half EF were tolerated, eHF was advanced to full strength (20 kcal/oz) and switched to preterm formula (24 kcal/oz) as soon as infants reached 1 kg of body weight. Advancements of EF were not defined using the “conservative” approach.

With “aggressive” nutrition, full PN was started with protein and fat on the first day of life. Enteral feeds were started using preterm formula (24 kcal/oz) on the first day of life. Clinicians were allowed to perform advancements of EF by a maximum of 20 ml/kg/d, which was considered to be safe in terms of triggering NEC (Victor & Simmer, 2005).

In both periods, mothers were encouraged to express breast milk, which was used instead of formula as soon as available. Fortification of mother’s milk was performed using a bovine multicomponent fortifier that was added as soon as 140 ml/kg/d (“conservative”) or 100 ml/kg/d (“aggressive”) of EF were reached. In both periods, the attending physicians individually prescribed EFs and PN. Physicians were trained to follow the new feeding guidelines by members of our nutrition team.

The components used for preparation of PN did not change within the study period. In particular the content of taurine in the amino acid solution (0.3 g/100 ml; Aminopaed 10%; Fresenius Kabi, Graz, Austria, EU) and the nature of lipid emulsion (exclusively soybean oil based; Intralipid 20%, Fresenius Kabi, Graz, Austria, EU) did not change. In both periods, urodeoxycholic acid was used for treatment—but not prevention (Gokmen et al., 2012)—of PNAC. Infants did not receive erythromycin (Aly et al., 2007; Gokmen et al., 2012; Ng et al., 2007) to reduce time until full enteral feeds. Fluconazole was used for fungal prophylaxis from 04/2006 (3 mg/kg twice a week) for a maximum of six weeks if a central line was present. All infants routinely received a peripherally inserted central catheter line for application of PN, either on the second day of life or on the seventh day of life if an umbilical venous line had been placed right after birth.

Study design and eligibility

The study was retrospective and observational. The primary aim was to evaluate the impact of an “aggressive” versus a “conservative” nutritional regimen on the incidence of PNAC in ELBW infants. The secondary aim was to compare postnatal growth. ELBW infants born within a two-year period of “conservative” (01/2005–12/2006) or “aggressive” (07/2007–06/2009) nutrition were screened for eligibility. Patients were excluded due to: (i) cholestasis at birth, (ii) diseases associated with cholestasis (i.e., inborn errors of metabolism, viral hepatitis, cystic fibrosis and any primary cholestatic liver diseases) that were diagnosed while hospitalized and (iii) early death or transfer to another hospital (i.e., within 28 days of life). Patients who died (but not those who were transferred) after 28 days of life were excluded from growth analysis, due to the risk of distortion of weight measurements by perimortal edema.

Data collection

Data were retrieved from the electronic patient’s charts (carevue®, Phillips Medical Systems, Eindhoven, The Netherlands, EU), the hospital’s PN prescription software (VIE-PNN (Horn et al., 2002), Medical University Vienna, Austria, EU) and discharge letters.

Demographic and basic clinical parameters

The full list of parameters is shown in Table 2. Prenatal steroids were defined as a full course (two doses) of betamethasone. Surfactant (200 mg/kg Curosurf®; Chiesi, Parma, Italia) was used in preterm infants with more than 35% oxygen on continuous positive airway pressure in the first 2 hours after birth. Starting from 2009, ELBW infants <27+6 weeks gestational age (GA) received surfactant prophylactically while spontaneously breathing as previously described (Klebermass-Schrehof et al., 2013). Percentiles and Z-scores of weight, head circumference and length were determined according to Fenton et al. (Fenton & Kim, 2013) using a spreadsheet based calculator (Fenton, 2013). Small for gestational age (SGA) was defined as birth weight <10th percentile. Infants of both study periods received neither probiotics nor lactoferrin for NEC/sepsis prophylaxis.

Table 2 Demographic and basic clinical parameters.

	Conservative (n = 77)	Aggressive (n = 85)	P	
Male sex	28 (36)a	47 (55)a	0.018	
Multiples	19 (25)a	28 (33)a	0.406	
Cesarean section	56 (73)a	63 (74)a	0.860	
Prenatal steroids	51 (66)a	66 (78)a	0.074	
PROM	44 (57)a	41 (48)a	0.164	
Preeclampsia	15 (19)a	8(9)a	0.054	
Apgar-5 min	8 [8–9]b,c	9 [8–9]b, d	0.339	
Surfactant	58 (75)a	66 (78)a	0.435	
Gestational age	26+5 [25+4–28+0]b	26+6 [25+2–28+0]b	0.795	
Birth weight (g)	795 [677–888]b	790 [660–890]b	0.527	
Percentile	34 [16–53]b	26 [13–45]b	0.331	
Z-score	−0.4 [−1–−0.1]b	−0.6 [−1.2–−0.1]b	0.455	
Birth length (cm)	33.4 [32.0–35.0]b	33.0 [31.0–35.0]b	0.559	
Percentile	29 [8–49]b	22 [9–49]b	0.717	
Z-score	−0.6 [−1.4–0.0]b	−0.8 [−1.4–−0.1]b	0.634	
Birth head circumference (cm)	23.8 [22.2–25.0]b	23.5 [22.3–24.5]b	0.634	
Percentile	32 [14–50]b	29 [13–50]b	0.611	
Z-score	−0.5 [−1.1–0.1]b	−0.6 [−1.1–0.0]b	0.596	
Birth body mass index	7.0 (6.5–7.6)	6.9 (6.3–7.7)	0.857	
Percentile	29.1 [9.0–58.3]	23.9 [4.9–52.3]	0.505	
<10th	20 (26.0)a	28 (33.0)a	0.390	
>90th	3 (3.9)a	5 (5.9)a	0.722	
Z-score	−0.6 [−1.4.–0.2]	−0.7 [−1.7–0.1]	0.583	
Small for Gestational Age	17 (22)a	17 (20)a	0.454	
Notes.

P values < 0.05 were considered statistically significant and are printed in bold letters.

Categorical data were tested using the Chi Square test, metric data were tested using students t test after inspection of histograms for normal distribution.

PROM premature rupture of membranes

HC head circumference

a number with percentage in parentheses.

b median and interquartile range.

c Data of 4 patients missing.

d Data of 12 patients missing.

Nutritional analysis in the first week of life

Data on prescribed PN (fluids, glucose, protein and lipids per kg/d), EF (fluids and total calories per kg/d) and total energy and fluids in infants of the “conservative” and “aggressive” period were expressed as the median during the first week of life (Table 3). Total days on PN were defined as days on any PN—with or without lipids. Furthermore infants with or without PNAC were compared for their parenteral and enteral nutritional intake within each population—together with important parameters on demography, morbidity and growth (Table 4).

Table 3 Nutritional analysis.

	Conservative (n = 77)	Aggressive (n = 85)	P	
Total days on PN	38 [27–53.5]	33 [23–48]	0.246	
Parenteral (1st week of life)	
Glucose (g/kg/d)	7.5 [6.6–8.1]	6.9 [6.3–7.6]	0.044	
Protein (g/kg/d)	1.9 [1.45–2.5]	2.8 [2.6–3.0]	0.000	
Lipids (g/kg/d)	1.2 [1.1–1.5]	1.9 [1.6–2.3]	0.000	
Fluids (ml/kg/d)	126.9 [114.5–139.6]	124.6 [108.8–144.5]	0.891	
Enteral (1st week of life)	
Energy (kcal/kg/d)	5.1 [2.9 –9.6]	19.7 [13.0–32.1]	0.000	
Fluids (ml/kg/d)	13.4 [8.2–21.7]	24.8 [16.5–40.3]	0.000	
Total (1st week of life)	
Energy (kcal/kg/d)	50.8 [44.2–55.7]	70.4 [60.3–77.8]	0.000	
Fluids (ml/kg/d)	142.9 [136.1–142.5]	152.9 [139.3–162.9]	0.000	
Notes.

Nutrition components, volumes and calories of ELBW infants in their first week of life were analyzed prior and after introduction of aggressive feedings. Data are presented as median and interquartile ranges. After inspection of histograms for normal distribution, total days on PN were tested using the Mann Whitney U test, all other parameters using the student’s t test.

P values < 0.05 were considered statistically significant and printed in bold letters.

PN parenteral nutrition

Table 4 Characteristics of infants with PNAC.

	PNAC	No PNAC	P	
NUTRITIONAL ANALYSIS 1st WEEK	
CONSERVATIVE PERIOD	
Parenteral	(n = 35)c	(n = 42)d		
Total days on PN	52 [39–63]	29 [22–38]	0.000	
Glucose (g/kg/d)	5.1 [4.5–5.5]	5.4 [4.6–6.0]	0.201	
Protein (g/kg/d)	1.8 [1.5–2.5]	1.9 [1.4–2.5]	0.883	
Lipids (g/kg/d)	1.1 [1.2–1.4]	1.2 [1.1 –1.6]	0.725	
Fluids (ml/kg/d)	130 [118–142]	122 [106–138]	0.141	
Enteral	
Energy (kcal/kg/week)	4.1 [2.6–7.0]	6.4 [3.4–12.9]	0.005	
Fluids (ml/kg/d)	8.0 [6.5–11.7]	13.7 [6.9–27.4]	0.009	
AGGRESSIVE PERIOD	
Parenteral	(n = 23)d	(n = 62)c		
Total days on PN	44 [30–81]	30 [22–43]	0.003	
Glucose (g/kg/d)	4.7 [4.5–5.6]	4.8 [4.4–5.3]	0.939	
Protein (g/kg/d)	2.9 [2.7–3.0]	2.5 [2.8–3.1]	0.404	
Lipids (g/kg/d)	1.8 [1.5–2.0]	1.9 [1.6–2.3]	0.177	
Fluids (ml/kg/d)	137 [117–149]	124 [101–138]	0.089	
Enteral	
Energy (kcal/kg/week)	15.5 [8.2–29.7]	20.1 [15.1–34.0]	0.073	
Fluids (ml/kg/d)	19.7 [10.3–37.1]	25.7 [19.0–42.7]	0.059	
Basic clinical parameters, morbidity and growth	
BOTH PERIODS	(n = 58)	(n = 104)		
Gestational age	26+1 [25+1–28+0]b	26+5 [25+3–27+6]b	0.440	
Male sex	30 (51.7)	45 (43.2)	0.327	
Birth weight, g	709 [627–823]b	824 [702–892]b	0.000	
Z-score	−0.75 [−1.4–−0.2]b	−0.47 [−0.9–+0.1]b	0.029	
Apgar 5 min	8 [7–9]b	8 [8–9]b	0.112	
Hospitalization, days	94 [67–109]b	71 [57–88]b	0.002	
Death	13 (22.4)a	7 (6.7)a	0.006	
Sepsis (culture proven)	27 (46.6)a	34 (32.7)a	0.092	
Necrotizing enterocolitis	12 (20.7)a	4 (3.8)a	0.001	
IVH grade 3/4	12 (20.7)a	12 (11.5)a	0.165	
BPD	25 (57.6)a,e	18 (17.3)a,f	0.000	
Medical treatment for PDA	46 (79.31)a	57 (54.8)a	0.002	
ROP (any stage)	21 (36.2)a,g	24 (23)a,h	0.039	
Weight at discharge	2174 [1931–2680]b	2200 [1777–2662]b	0.118	
Notes.

Data were tested using the Chi Square test for categorical data and the student’s t test for continuous variables (except for total days on PN using Mann Whitney U Test).

P values < 0.05 were considered statistically significant and printed in bold letters.

BPD Bronchopulmonary Dysplasia

IVH Intraventricular Hemorrhage

ROP Retinopathy of Prematurity

a number with percentage in parentheses.

b median and interquartile range.

c Missing data: Data of 1 infants missing.

d Missing data: Data of 3 infants missing.

e Missing data: Data of 13 infants missing.

f Missing data: Data of 26 infants missing.

g Missing data: Data of 9 infants missing.

h Missing data: Data of 10 infants missing.

Neonatal outcome: PNAC and other neonatal morbidities

Neonatal morbidities including the primary outcome are presented in Table 5. Laboratory analyses including bilirubin with fractions were performed at least weekly as long as PN was required, afterwards every 7–14 days and at discharge. Conjugated bilirubin was measured by spectrophotometric quantitation (Vitros Chemistry System, Ortho Clinical Diagnostics, Raritan, NJ, USA). Parenteral nutrition associated cholestasis was defined as conjugated bilirubin >1.5 mg/dl (25 µmol/l) measured at two consecutive occasions (Bines, 2004) which corresponds closely to >2 mg/dl direct bilirubin (Ye et al., 2015). Conjugated bilirubin levels were used as they are more accurate compared to direct bilirubin (Doumas & Wu, 1991), in particular when bilirubin levels rise and persist over a longer period of time (Ye et al., 2015). NEC was diagnosed either clinically (modified Bell’s stage ≥ IIa (Walsh & Kliegman, 1986)) or after surgical exploration. Focal intestinal perforation was diagnosed after surgical exploration if a single perforation occurred in an otherwise healthy bowel. Intraventricular hemorrhage (IVH) grade 3/4 and cystic periventricular leucomalacia were diagnosed by ultrasound (De Vries, Eken & Dubowitz, 1992; Volpe, 1989) that was performed every 7–14 days. Bronchopulmonary dysplasia (BPD) was defined as need for supplementary oxygen after 36+0 weeks of GA. Pharmacological treatment of persistent ductus arteriosus (PDA) using ibuprofen was performed based on its hemodynamic significance in ultrasound and cardiorespiratory stability. Screening for retinopathy of prematurity (ROP) by indirect ophthalmoscopy was started at 5 weeks of chronological age.

Table 5 Neonatal outcome—morbidity.

	Conservative (n = 77)	missing	Aggressive (n = 85)	missing	P	
Hospitalization, days	77 [61–98]b		78 [58–100]b		0.710	
Death	9 (11.7)a		11 (13.0)a		1.000	
PNAC	35 (45.5)a		23 (27.0)a		0.021	
Onset (day of life)	34 [20–45]b		30 [12–39] b		0.460	
Peak conjugated bilirubin	5.0 [3.2–8.4]b		4.7 [3.3–7.4]b		0.851	
Mortality	8/35 (22.9)a		5/23 (21.7)a		1.000	
Sepsis (culture proven)	36 (46.7)a		25 (29.4)a		0.024	
Necrotizing enterocolitis	8 (10.4)a		8 (9.4)a		1.000	
Focal intestinal perforation	1 (1.3)a		1 (1.2)a		0.726	
IVH grade 3/4	14 (18.1)a		10 (11.8)a		0.275	
Surgery (any)	26 (33.8)a		35 (41.2)a		0.209	
Surgery (GI)	11 (14.3)a		8 (9.4)a		0.464	
NEC and/or GI surgery	12 (15.6)a		11 (12.9)a		0.658	
Cystic PVL	3 (3.9)a		1 (1.2)a		0.274	
BPD	19 (24.6)a	16 (20.7)a	24 (28.2)a	23 (27.0)a	0.451	
Steroids for BPD	16 (20.8)a		10 (11.8)a		0.137	
Medical treatment for PDA	44 (57.1)a		59 (69.4)a		0.141	
ROP (any stage)	18 (23.3)a	8 (10.4)a	27 (31.8)a	11 (12.9)a	0.209	
Notes.

Univariate analysis of neonatal outcome parameters. Categorical data were tested using the Chi Square test; metric data were tested using the student’s t test.

P values < 0.05 were considered statistically significant and printed in bold letters.

a number with percentage in parentheses.

b median and interquartile range.

PNAC parenteral nutrition associated cholestasis

IVH intraventricular hemorrhage

PVL periventricular leucomalacia

BPD bronchopulmonary dysplasia

PDA persistent ductus arteriosus

ROP retinopathy of prematurity

GI gastrointestinal

A multivariate analysis on the effect of “aggressive” nutrition on the incidence of PNAC corrected for confounders (see statistics) is presented in Table 6.

Table 6 Multivariate analysis on aggressive nutrition and the risk for PNAC.

	Adjusted OR	CI	P	
Aggressive Nutrition	0.273	0.115–0.645	0.003	
Male Sex	2.163	0.951–4.920	0.066	
Sepsis	1.097	0.468–2.573	0.832	
NEC and/or GI surgery	1.214	0.468 –2.573	0.764	
Birth Weight	0.998	0.994–1.002	0.325	
Total days on PN	1.054	1.028–1.081	0.000	
Z-score of birth weight	0.415	0.232–0.734	0.003	
Notes.

Binary logistic regression analysis showing the corrected odds for aggressive nutrition and PNAC correcting for the co-variates male sex, sepsis, necrotizing enterocolitis (NEC) and/or gastrointestinal (GI) surgery, birth weight, duration of parenteral nutrition (PN) and the degree of growth retardation at birth.

OR odds ratio

CI 95% confidence interval

P values < 0.05 were considered statistically significant and printed in bold letters.

Neonatal outcome: growth

Body measures (body weight, crown-heel length and head circumference) are presented together with Z-scores in standard deviations (SD) and the body mass index (BMI, weight in g *10/ cm2) with respective percentiles, the percentage of infants <10th and >90th BMI percentile as well as Z-scores at birth (Table 2) and at discharge (Table 7). The BMI percentiles and Z-scores were generated by the authors of the publication by Olsen et al. (2015) - using their data set of intrauterine BMI-for-age growth curves. For infants discharged later than 42 weeks of GA, BMI percentiles could not be calculated and data is therefore missing. The difference of Z-scores from birth to discharge is reported in Table 7. Growth failure was categorized according to Shah et al. (Shah et al., 2006) as having no (Δ Z-score higher than −1 SD), mild (Δ Z-score between −1 to −2 SD) or severe (Δ Z-score below −2 SD) growth failure and is expressed as the percentage of discharged infants (Fig. 1).

Table 7 Neonatal outcome–Growth characteristics at discharge.

	Conservative (n = 67)	Aggressive (n = 74)	P	
Postmenstrual age	38+2 [36+5–39+5]a	38+2 [36+6–39+6]a	0.991	
Weight (g)	2050 [1802–2235]a	2461 [2020–2783]a	0.000	
Z-score (SD)	−2.34 [−2.85–−1.74]a	−1.38 [−1.9–−0.74]a	0.000	
ΔZ-score (SD)	−1.7 [−2.16–−1.27]a	−0.82 [−1.16–−0.32]a	0.000	
Length (cm)	43 [41–45]a	44 [42–46]a	0.017	
Z-score (SD)	−2.8 [−3.9–−2.3]a	−2.1 [−2.85–−1.4]a	0.000	
ΔZ-score (SD)	−2.1 [−2.85–−1.60]a	−1.35 [−1.92–−0.58]a	0.000	
Head circumference (cm)	30.6 [29.5–31.5]a	31.6 [30.3–33.0]a	0.007	
Z-score (SD)	−2.1 [−2.8–−1.5]a	−1.4 [−1.82–−0.8]a	0.000	
ΔZ-score (SD)	−1.5 [−2.03–−1.08]a	−0.8 [−1.63–−0.15]a	0.001	
Body mass index	11.1 (10.3–11.9)c	12.4 (11.1–13.7)	0.000	
Percentile	11.8 (3.8–33.3)d	41.2 (21.4–67.9)e	0.000	
<10th	25 (43.8)b,d	6 (9.2)b,e	0.000	
>90th	2 (2.6)b,d	3 (4.6)b,e	0.562	
Z-Score (SD)	−1.2 (−1.8–−0.5)e	−0.2 (−0.8–0.5)e	0.000	
ΔZ-score (SD)	−0.4 (−1.1–0.2)e	0.7 (−0.4–1.4)e	0.000	
Notes.

Data were tested using students t test after inspection of histograms for normal distribution

P values < 0.05 were considered statistically significant and printed in bold letters

a median and interquartile range.

b number with percentage in parentheses.

c Data of 2 infants missing.

d Data of 10 infants missing.

e Data of 9 infants missing.

GA gestational age

SD standard deviation

Figure 1 Postnatal growth failure of weight, head circumference and length in extremely low birth weight infants nourished using a “conservative” or “aggressive” nutritional regimen.

Categorisation of postnatal growth failure was performed according to the difference in Z-score (Δ Z-score in standard deviations, SD) from birth to discharge as “no” (white bars; Δ Z-score higher than −1 SD), “mild” (grey bars, Δ Z-score between −1 and −2 SD) or “severe” (black bars, Δ Z-score below −2 SD). Data are presented as cases/total infants with percentages in parentheses. The difference in the distribution of infants with growth failure (“mild” and “severe”) and those without (“no”) was tested using the Chi Square test. P < 0.05 was considered statistically siginficant.

Statistics

Statistical analysis was performed using the χ2 test for categorical data; student’s t and Mann Whitney U test were used for continuous data depending on their normal distribution as appropriate. For statistical analysis of BMI percentiles from birth to discharge, the paired student’s t test was used. The primary outcome PNAC was analyzed in a model of binary logistic regression to investigate the influence of “aggressive” nutrition on PNAC corrected for relevant confounding factors of cholestasis (NEC and/or gastrointestinal (GI) surgery, culture proven sepsis, birth weight, Z-score of birth weight, time on PN and male sex) and the adjusted odds ratio is reported (Table 6). P-values < 0.05 were regarded as statistically significant. Data represent the median and interquartile range if not otherwise indicated.

Ethics and registration

Due to the retrospective nature of the study, parental informed consent was not considered necessary by the institute’s ethics board that approved the study design. The study was concomitantly registered at ClinicalTrials.gov as a retrospective cohort study (NCT01164878).

Results

Screening

A total of 296 ELBW infants were born within the study period (“conservative” 01/2005–01/2006 (n = 153), “aggressive” 07/2007–06/2009 (n = 143)). Of 296 infants screened, 162 infants (“conservative”: n = 77, “aggressive”: n = 85) were eligible. Reasons for exclusion were death <28 days of life (“conservative”: n = 54; “aggressive”: n = 45), transfer <28 days of life (“conservative”: n = 21; “aggressive”: n = 13) and congenital cytomegalovirus infection (“conservative”: n = 1).

Demographic and basic clinical parameters

Demographic and basic clinical parameters were not significantly different except for “male sex” (significantly higher proportion in the “conservative” group, Table 2).

Nutritional analysis first week of life

ELBW infants nourished using the “aggressive” regimen received significantly higher amounts of parenteral amino acids and lipids, but significantly less glucose compared to the “conservative” regimen (Table 3). The amount of parenteral fluids was equal, whereas enteral feeding volumes were almost twice as high using “aggressive” nutrition. Enteral nutrition provided four times more calories in the “aggressive” period and consecutively total energy was significantly higher. Time on PN was 5 days shorter using “aggressive” nutrition (statistically not significant).

Neonatal outcome—morbidities

The incidence of the primary outcome PNAC was significantly lower in the period of “aggressive” (27.0%) compared to “conservative” (45.5%) nutrition (Table 5). Furthermore, culture proven sepsis occurred significantly less often in the “aggressive” period. Other parameters were not significantly different. In a multivariate analysis on the effect of “aggressive” nutrition corrected for confounding variables of cholestasis the effect of “aggressive” nutrition on the reduction of PNAC was statistically significant (adjusted OR 0.275, P < 0.01; Table 6).

Neonatal outcome—growth

Extremely low birth weight infants on “aggressive” nutrition were discharged at similar GA but with significantly higher body weight (+400 g), length (+1 cm) and head circumference (+1 cm). Postnatal growth faltering was significantly decreased (Table 7) and the percentage of infants diagnosed with “mild” or “severe” growth restriction was markedly reduced (Fig. 1). The positive effect on postnatal growth restriction was most pronounced for body weight, followed by head circumference, and body length (Table 7 and Fig. 1). At discharge, the median BMI in the “conservative” group was significantly lower compared to the “aggressive” group, while there was no difference between the two groups at birth (Table 2). The median BMI percentile at birth was similar in the “conservative” and “aggressive” group. Infants of the “conservative” group showed a significant loss of BMI percentiles from birth to discharge (from 29th to 11th percentile, P = 0.004), while infants of the “aggressive” group significantly gained BMI percentiles (from 23rd to 41st percentile, P = 0.004; Tables 2 and 7). The proportion of infants discharged below the 10th BMI percentile was significantly lower after switching to “aggressive” nutrition, while the percentage of infants discharged over the 90th BMI percentile did not increase (Table 7).

Characterization of infants with PNAC

ELBW infants without PNAC (Table 4) received higher amounts of enteral energy and fluids and were weaned from PN significantly faster. Infants with PNAC of both periods were born with a significantly lower birth weight and also Z-score of birth weight, displayed a higher mortality and morbidity (NEC, BPD, PDA requiring medial treatment, ROP) and were significantly longer hospitalized compared to infants without PNAC.

Discussion

Implementation of “aggressive” nutrition for ELBW infants was associated with a reduced incidence of PNAC and improved weight, head circumference, length and BMI at discharge.

After implementing new nutrition guidelines at our unit, nutrition of ELBW infants was performed more “aggressively.” This included to introduce parenteral amino acids and lipids earlier and at higher amounts, as well as to start enteral nutrition immediately after birth with feeding advancements up to 20 ml/kg/day. To evaluate whether the new guidelines were put into practice, we analyzed the nutritional intake in the first week of life. Here, we found that ELBW infants received significantly more parenteral amino acids and lipids, but less glucose (Table 3). The lower glucose supply was unexpected as glucose was also started at increased amounts using the “aggressive” regimen (Table 1). It thus seems that parenteral glucose needs were lowered, most probably because EFs were started earlier and increased faster. Despite accelerating early feeding advancements, the total time on PN was not significantly shortened (−5 days, P = 0.246, Table 3). We can only speculate whether caution to avoid NEC or feeding intolerance slowed down care keepers in weaning infants from PN. Maybe feeding advancements were simply not defined rigorously enough, but the data on safety in ELBW infants are still too limited (Morgan, Young & Mcguire, 2015) to justify a fixed feeding advancement. Finally, it may as well represent a power problem and a significant difference might have been detectable if a larger group of infants had been recruited.

We were concerned about a negative impact of “aggressive” nutrition on PNAC—due to the early increase of amino acids (Steinbach et al., 2008) and lipids (Carter et al., 2007; Clayton et al., 1993; Kurvinen et al., 2012) in PN—especially since one randomized trial of “aggressive” PN showed a significant increase of total bilirubin in very low birth weight (VLBW, <1,500 g birth weight) infants using “aggressive” PN (Ibrahim et al., 2004). Also the influence of initiating EFs with full strength preterm formula instead of diluted hydrolyzed feedings and earlier fortification on the incidence of PNAC, feeding tolerance and NEC (Pearson, Johnson & Leaf, 2013) are ill-defined. It was therefore reassuring to find that the incidence of PNAC was even significantly reduced (Tables 5 and 6) after modifying our feeding regimen. The severity of PNAC did not change as mortality and the highest conjugated bilirubin levels were relatively similar (“conservative” 5.0 vs. “aggressive” 4.7 mg/dl, Table 5). The severity of PNAC in our study infants was generally comparable to another study in VLBW infants by Costa et al. using a rather “conservative” nutritional regimen (4.9–5.2 mg/dl direct bilirubin (Costa et al., 2010)). Concerning the tolerance of more aggressive enteral nutrition, we observed a stable NEC rate between groups and speculate that the faster progression of EFs with higher osmolality had no negative impact. Sepsis is an important trigger for cholestasis and culture proven sepsis was significantly lower in the period of “aggressive” nutrition. Interestingly, the patient-related risk factors for sepsis (time on PN, BPD, NEC and birth weight (Stoll et al., 2002)) did not significantly differ between the two periods. A relation of “aggressive” nutrition to reduction of sepsis can therefore not be excluded, but the effect is more likely caused by hygiene improvements (Legeay et al., 2015) at our unit.

To exclude that other factors besides a more “aggressive” nutritional regimen–especially the lower sepsis rate-reduced PNAC, we performed a multivariate analysis (Table 6) including sepsis (Beale et al., 1979; Beath et al., 1996; Drongowski & Coran, 1989), low birth weight (Beale et al., 1979), NEC/GI surgery (Moss, Das & Raffensperger, 1996; Veenstra et al., 2014), duration of PN (Zambrano et al., 2004), male sex (Albers et al., 2002; Bines, 2004) and Z-score of birth weight. While most factors are well established, the data on the significance of intrauterine growth restriction are controversial (Costa et al., 2010; Robinson & Ehrenkranz, 2008). As infants with PNAC showed significantly lower Z-scores of birth weight in our study (Table 4), the degree of growth retardation seemed relevant in our cohort and was therefore included into analysis. We did not correct for duration of hospitalization and neonatal morbidities like BPD, ROP or PDA—that had a significantly higher prevalence in infants with PNAC (Table 4)—as these factors are no risk factors for PNAC, but rather linked to PNAC via low birth weight. We further did not include fluconazole prophylaxis, as its association with PNAC (Aghai et al., 2006) was not confirmed by randomized trials (Ericson et al., 2016). After considering all confounders the effect of “aggressive” nutrition on the reduction of PNAC still remained statistically significant (adjusted OR 0.275, Table 6). Due to the retrospective nature of the study, it is still possible that there may be other confounding factors not considered within the study.

As PNAC remained significantly reduced after correction for non-nutritional confounders it seems most plausible, that nutritional factors—modified by “aggressive” nutrition—were involved. In view of the lack of literature on “aggressive” or “conservative” nutrition of ELBW infants and PNAC incidence, the putative mechanisms deserve some more attention. “Aggressive” nutrition contained a bundle of changes to our feeding regimen (Table 1) that are related to PNAC. One explanation how PNAC is reduced by “aggressive” nutrition could be a reduction of the time on PN (Zambrano et al., 2004) and thus a reduced exposure to soybean oil based lipids (Carter et al., 2007; Clayton et al., 1993; Kurvinen et al., 2012). Furthermore enteral nutrition right after birth (Drongowski & Coran, 1989; Yip, Lim & Tan, 1990) is enhanced, thus stimulating bile flow. In our cohort, infants with PNAC generally received PN significantly longer (Table 4), which also implicates a higher cumulative amount of parenteral lipids (Nayrouz & Amin, 2014). As significantly more parenteral lipids were applied in the first week of life using “aggressive” nutrition (Table 3), it seems questionable whether a reduced lipid exposure could be part of the observed reduction of PNAC after changing our nutritional regimen. Nevertheless, there was also a trend towards a reduced time on PN and infants still may have received a lower cumulative amount of soybean oil based lipids. It is however a limitation of our study that our nutritional analysis did not extend beyond the first week of life and we cannot answer this question in our setting. On the other hand, early enteral nutrition was significantly enhanced after establishing “aggressive” nutrition, with almost twice as much EFs applied in the first week of life (Table 3) and early enteral nutrition had a positive influence against PNAC (Table 4). It thus seems that early stimulation of bile flow by “aggressive” enteral nutrition was a relevant factor for prevention of PNAC–while we cannot exclude a role of a reduced cumulative amount of lipids.

Postnatal growth restriction was common at our unit as elsewhere (Ehrenkranz et al., 1999). By implementing “aggressive” nutrition, we could markedly improve weight gain, head and also linear growth of ELBW infants (Table 7). In particular, severe postnatal growth faltering of weight (<−2 SD)—which is associated with unfavorable neurodevelopment (Shah et al., 2006)—could almost be completely avoided (Fig. 1). The aim of nutritional care of preterm infants is defined as reaching a postnatal growth that is comparable to the healthy human fetus (American Academy of Pediatrics Committee on Nutrition, 1985). In this respect, the proportion of ELBW infants without faltering of weight gain increased from 15% to 65% while the median Z-score loss of body weight improved from −1.7 to −0.82. Interestingly, a Z-score loss of −0.8 corresponds to what was recently defined as natural weight loss by extracellular contraction in a multicenter cohort of healthy VLBW infants (Rochow et al., 2016), which is therefore encouraging. However, the improvements in normal head growth (from 28% to 55%) and linear growth (from 15% to 35%) at our unit were less pronounced. The finding that weight and head circumference are more susceptible to nutritional intervention than linear growth are quite similar to Roggero et al. (2012) who reported improved weight and head circumference, but not length after introducing a comparable nutritional regimen. Such “disproportionate” postnatal growth characterized by primarily stunting (Ramel et al., 2012) raises concerns about body composition and potential adverse metabolic effects (Yeung, 2006). In this context, we found that the BMI at discharge was significantly higher using “aggressive” compared to “conservative” nutrition (Table 7). Recently, Olsen et al. (2015) published fetal BMI percentiles that enable to evaluate the relation of lean and fat mass of hospitalized preterm infants. Roughly a quarter of ELBW infants in our study were born with a BMI below the 10th percentile (Table 2). Calculating the BMI percentiles we found that infants of the “conservative” period lost a BMI trajectory from birth (29th percentile) to discharge (11th percentile) and that the proportion of infants discharged below the 10th BMI percentile doubled to almost 50 percent (Tables 2 and 7). On the contrary, infants of the “aggressive” group gained one trajectory from the 24th to the 41st BMI percentile and the proportion discharged below the 10th BMI percentile dropped below 10 percent. Concerning a possible adverse health effect due to overfeeding, there was no increase in the proportion of infants that were discharged with a BMI over the 90th BMI percentile (Table 7). Thus, growth generally became more proportionate using the “aggressive” regimen and—given the correlation of BMI with fat mass in preterm infants (Cooke & Griffin, 2009)—these findings suggest that exaggerated fat accumulation did not happen. Unfortunately we cannot present any data on direct measurement of body composition due to the retrospective nature of the study. Future prospective studies that aim at further improving especially linear growth of preterm infants should probably not be performed without direct measurement of body composition (Rice & Valentine, 2015) to evaluate potentially unhealthy growth.

Conclusion

Our study showed that a combined approach of “aggressive” parenteral and enteral nutrition significantly reduced PNAC while improving postnatal growth in a proportionate manner.

Supplemental Information

Data S1 SPSS file of analyzed data

Click here for additional data file.

The authors would like to thank Louise Lawson from the Department of Statistics and Analytical Sciences at the Kennesaw State University (Georgia, USA) for calculating the individual BMI percentiles and Z-scores and Irene Olsen from the University of Pennsylvania (Pennsylvania, USA) for her permission to use the data.

Additional Information and Declarations

Competing Interests

Author Contributions

Human Ethics

Data Availability

The authors declare there are no competing interests.

Andreas Repa conceived and designed the experiments, performed the experiments, analyzed the data, wrote the paper, prepared figures and/or tables, reviewed drafts of the paper.

Ruth Lochmann conceived and designed the experiments, performed the experiments, analyzed the data, prepared figures and/or tables.

Lukas Unterasinger conceived and designed the experiments, performed the experiments, analyzed the data, contributed reagents/materials/analysis tools, reviewed drafts of the paper.

Michael Weber analyzed the data, contributed reagents/materials/analysis tools, reviewed drafts of the paper.

Angelika Berger conceived and designed the experiments, reviewed drafts of the paper.

Nadja Haiden conceived and designed the experiments, wrote the paper, reviewed drafts of the paper.

The following information was supplied relating to ethical approvals (i.e., approving body and any reference numbers):

Ethikkommission der Medizinischen Universität Wien

Borschkegasse 8b/E06

1090 Vienna

Austria

Ethics board number EK 2010/061.

The following information was supplied regarding data availability:

The raw data has been supplied as Data S1.

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
