# Peer review of "Aggressive nutrition in extremely low birth weight infants: impact on parenteral nutrition associated cholestasis and growth"

_PeerJ, doi:10.7717/peerj.2483_

## Round 0.1 · original submission · Major Revisions

Please make sure to address the excellent comments by the two reviewers in your revision.

Reviewer 1 ·

Basic reporting

Intro and background are generally relevant, with good language.
In the abstract though, the Results section, growth failure is defined as only regarding bodyweight, which is prone to misunderstanding, since three growth variables are reported.

The literature is not always adequately referenced throughout the manuscript. Too often review articles and textbooks are referred to (Klein et al 2010, Carter 2007, Ehrenkrantz 2007, Bines 2004 for example). References to original work should be provided instead, to support the various statements.

Experimental design

The submission is within the scope of the journal.

The submission clearly defines a relevant research question. A knowledge gap is identified, and the study attempts to fill that gap.

The study is retrospective and observational, which is acceptable when there are no prospective studies available. The study design has the limitation that conclusions on causality cannot be drawn.


The definition of PNAC, >1.5mg/dL used in the study is somewhat unusual. Many other studies use a definition of conjugated bilirubin >2mg/dL. Do the authors have any
rationale or a better reference than the one provided?

Validity of the findings

In general this is a very interesting and well-performed study given the limitations of the study design comparing two different time periods.

Some issues have been identified in the review:

In all tables the statistical method used to produce each p-value should be presented in the legend.

PNAC
The authors should present / describe data on cholestasis among infants with PNAC to show the reader that it is clinically significant cholestasis, for example max conj. bilirubin, age of max bilirubin, duration of cholestasis etc as these variables are available in the raw data file provided.
The authors should also describe how often conjugated bilirubin is tested in these ELBW infants.
A univariate comparison of variables between PNAC infants and non-cholestatic ones within the same population (aggressive or conservative) should also be presented, since for example, the authors state that the cholestatic infants have more growth retardation at birth, do not show the data, but include it in their multivariate analysis.


Figure 1 is relevant, but the statistical method used is questionable. There is a clear improvement of delta-Z-scores between the time periods studied, as seen in the figure. The chi-square test using three categories cannot really prove that there has been an improvement over time, only that there is a significantly different distribution. For example, in the “length”-figure, the proportion of mild postnatal growth failure has actually increased in the aggressive time period.
A more appropriate use of the chi-square test would be to use two categories, for example no growth failure vs growth failure, or mild or no growth failure vs severe growth failure for example.

Table 2: It should be made clear that percentile and delta-Z-score are related to the prior growth variable, for example birth weight, by using for example indention

Table 6: In the manuscript body regarding this table, in the results section, regarding Neonatal outcome – Growth, the reasoning regarding BMI percentiles is hard to follow since data is not shown, and should be omitted, or the data must be shown.
In table 6 there seems to be a misprint regarding note (a) and (b) in the table and in the legend.

Table 7: Since PNAC was more common in the “conservative” time period, it is not surprising that they on average received less enteral nutrition and less parenteral lipid infusion in the first week of life. Because of the design of the study where two different time periods are compared, the nutrition among infants should rather be compared within each population between cholestatic and non-cholestatic infants, which would be more informative.

In the discussion, the authors speculate why total duration of PN did not decrease significantly. They should mention that this may very well be a power problem, and with a larger study there may have been a significant difference.

Authors conclude that a stable NEC rate between the time periods indicate that the faster progression of EF with higher osmolality had no negative impact. This is a reasonable and interesting speculation, but should be presented as such and not as a conclusion on causality.

The results on differences in growth between the time periods are very interesting, but the study design prevents the authors from drawing conclusions on causality.
The authors do not discuss the possibility that there may be other confounding factors not evaluated within the study that also could affect the results.

Reviewer 2 ·

Basic reporting

minor comments
1. Page 8, line 139 : with our without -> with or without

2. Page 12, line 207
The incidence of the primary outcome PNAC was significantly lower in the period of “aggressive” compared to “conservative” nutrition.
 The incidence of the primary outcome, PNAC was significantly low in the period of “aggressive” compared to “conservative” nutrition.

Experimental design

1. PNAC, commonly defined as direct bilirubin ≥ 2.0mg/dL in many previous reports. Is there any special reason for defining PNAC as direct bilirubin ≥1.5mg/dL?

2. The authors compared the nutritional intake only for the first week of life. Previous reports had compared the nutritional volume during the whole time period of PN or at least first 2-4 weeks of PN to determine the effects of amount and composition of PN on PNAC.

3. Death should be included as an outcome.

4. Comment on timing of full enteral feeding in both groups.

5. Table 5 showed multivariate analysis of risk factors of PNAC. The others which is known as risk factors for PNAC was not considered for analysis? For example, fluconazole prophylaxis, erythromycin use, UDCA therapy, GI surgery, BPD, longer duration of ventilator care or hospitalization….

Validity of the findings

The main conclusion of this article is that "Aggressive nutrition was assoicated with a significnat decrease of PNAC in ELBW infants."
Sure enough, the results showed this. However, this is a retrospective study. So, it is just observational data.
It seems lacking of discussion how aggressive nutrition decrease the risk of PNAC.

In general, “aggressive nutrition” enables to take less time to reach full enteral feeding, results in shortening duration of PN, and thereby results in decreasing the risk of catheter-related complications including sepsis. By this process, aggressive nutrition has a preventive effect of the development of PNAC. However, there was no difference in PN duration between aggressive and conservative groups. Authors commented that early stimulation of bile flow by ‘aggressive” enteral nutrition prevented PNAC.
Aggressive PN may increases the risk of PNAC. Aggressive enteral nutrition decreases the risk of PNAC. Aggressive nutritional support improves the postnatal growth in ELBW infants.

What is the point that the authors want to emphasize?
I understand that the authors state that Aggressive nutrition improved postnatal growth. And besides, it did reduce the risk of PNAC.
More discuss about this part.

Additional comments

With increasing survival of more preterm babies, PNAC has becoming an important issue in neonatal intensive care unit. This report addressed to the effects of aggressive nutrition on PNAC and postnatal growth in ELBW infants. It is already well-established that aggressive nutrition improves postnatal weight gain in preterm infants. However, as the comments of authors, there is lack of literatures about the association of aggressive nutrition and PNAC in ELBW infants.

The paper could be strengthened by addressing the associations of aggressive nutrition and PNAC in ELBW infants.

---

## Round 0.2 · accepted · Accept

The suggestions of the two reviewers have been addressed carefully, thank you.